# UPLC-MS/MS Method for Simultaneous Determination of Valnemulin and Its Metabolites in Crucian Carp: In Vivo Metabolism and Tissue Distribution Analyses

**DOI:** 10.3390/molecules28145430

**Published:** 2023-07-15

**Authors:** Qiyu Yang, Xiaojun Zhang, Qianfeng Wang, Yaqian Zhong, Wenjing Liu

**Affiliations:** 1School of Food and Pharmacy, Zhejiang Ocean University, Zhoushan 316022, China; yangqiyu@zjou.edu.cn (Q.Y.); wangqianfeng@zjou.edu.cn (Q.W.); zhongyaqian@zjou.edu.cn (Y.Z.); liuwenjing004006@163.com (W.L.); 2Key Lab of Mariculture and Enhancement of Zhejiang Province, Zhejiang Marine Fisheries Research Institute, Zhoushan 316021, China

**Keywords:** valnemulin, detection, crucian carp, metabolites

## Abstract

Valnemulin (VML) is a semi-synthetic pleuromutilin derivative widely used to treat animal bacterial diseases. However, no study has comprehensively evaluated VML metabolism in aquatic animals, including crucian carp. This study aimed to investigate VML metabolism in crucian carp. VML metabolites in crucian carp were quantified via intraperitoneal injection and analyzed via ultra-performance liquid chromatography-tandem mass spectrometry (UPLC-MS/MS). Three VML metabolites were detected in crucian carp via ultra-performance liquid chromatography-tandem quadrupole and time-of-flight mass spectrometry (UPLC-Q-TOF/MS) structural analysis. The enrichment and metabolism rules of the metabolites were summarized based on tissue distribution and concentration changes of the three metabolites. The metabolites were mainly found in the liver at 0.1 h after VML injection. The levels of the metabolites were abundant in the bile from 4 h to 12 h and in the skin after 72 h. The levels of the metabolites in the bile first increased, then decreased. The metabolism in the liver was completed at 72 h. The metabolites were detected in the skin following a 72 h period, which increased with time.

## 1. Introduction

Valnemulin (VML) is a semi-synthetic pleuromutilin antibiotic derivative which induces antimicrobial activities against mycoplasma and brachyspira by inhibiting bacterial protein synthesis [1,2]. VML has been widely used to treat swine dysentery, swine enzootic pneumonia, and mycoplasma infection in poultry and cattle [3]. Furthermore, VML is widely used for prophylactic and somatotrophic purposes in livestock. However, VML is associated with human health risks due to the toxicity of the veterinary antibiotic residues used in food-producing animals. VML residues may cause other bacteria in the human body to mutate into drug-resistant bacteria, affecting the efficacy of antimicrobial agents. The emergence of these resistance genes in animals poses a potential threat to human medicine and increases the difficulty of prevention and treatment of human diseases [4]. The accumulation of drug residues and metabolites in eggs, meat, and milk can affect human health [5,6]. According to the European Medicines Agency, the maximum residue limits (MRLs) for muscle, kidney, and liver are 50 μg/kg, 100 μg/kg, and 500 μg/kg, respectively. Furthermore, the Positive List System for Agriculture Chemical Residues in Food of Japan recommended that the MRL for swine muscle, fat, kidney, and liver be 50 μg/kg [7].

In recent decades, VML has been extensively studied in terrestrial animals, including rats, chickens, ducks, dogs, and pigs [8], with VML widely distributed in animal tissues [9]. In recent studies, pharmacokinetic and bioavailability studies of VML were conducted in healthy Muscovy ducks [10] and layer chickens [11] after intravenous (IV), intramuscular (IM), and oral administration of certain doses. The detailed pharmacokinetic profiles showed that this drug is widely distributed and rapidly eliminated; however, it has a low bioavailability, indicating that VML is likely to be a favorable choice in clinical practice. In addition, VML can be rapidly excreted, mostly via bile and feces [12]. Nonetheless, no study has comprehensively evaluated VML metabolism in aquatic animals. In our research team, Zong et al. [13] established a novel analytical method for the determination of VML residues in aquatic products. At present, a small number of applications of VML have been found in aquaculture in China. Therefore, VML metabolism in aquatic animals should be evaluated to assess metabolites and their metabolic and distribution status.

Crucian carp is one of the most important freshwater fish species in China and a classic animal model for various toxicological studies. HPLC [14], LC-MS/MS [15,16], LC-Q-TOF [8], and monoclonal antibody-based paper sensors are the major methods used for VML detection in animal tissues [17]. However, these methods are time-consuming due to the complex sample preparation and sample matrix interference and have low sensitivity, and thus are not suitable for VML detection in fish tissues. In this work, VML and its metabolites were detected in tissues of crucian carp using the UPLC-MS/MS method. Sample preparation and detection procedures were optimized to achieve high throughput and sensitivity. The metabolism and distribution of VML and its metabolites in crucian carp were also assessed via the UPLC-MS/MS method. This study provides the basis for exploring the residue pattern of VML and the control of VML in aquaculture.

## 2. Results and Discussion

### 2.1. Method Validation

In this study, the LOD and LOQ of VML in all tissues was 0.03 μg/kg and 0.1 μg/kg, respectively. The percentage of the response value of the same target and the response value of VML in pure solution was determined by adding a certain amount of target VML in the treated blank matrix, obtaining a matrix effect (ME) value of about 92.0%, indicating that the matrix effect was insignificant. The solvent standard calibration curve had a good linear correlation over the range of 5–1000 μg/L with the weighted r^2^ of 0.997356, indicating that the method had a high selectivity with no interferences (regression equation: y = 2.96825x + 1.00461). The average recovery rates of VML in crucian carp muscle, liver, and intestine (Table 1), were about 96.72%~113.78%, 99.23%~115.43%, and 80.35%~111.41%, respectively (relative standard deviation; 1.02%~3.68%, 2.60%~7.06%, and 1.68%~7.24%, respectively). The relative standard deviation between batches was about 2.26%~9.10%. Song et al. [18] found that the average recovery rate of darfloxacin in Yellow River crucian carp is about 85.12%~95.46%. Furthermore, Na et al. [17] found that the intra-batch and inter-batch VML recoveries in crucian carp are about 81.17–102.01% (highest coefficients of variation (CV); 9.81%), and 80.66–99.11% (highest CV; 10.79%), respectively. Li et al. [7] found that the mean VML recoveries in fortified swine tissues and bovine tissues are about 93.4–104.3% (CV; 3.3–8.7%) and 94.7–103.5% (CV; 4.1–10.7%), respectively. The VML recovery rate in this study was higher than those of the above two studies, indicating that the method is suitable for the determination of VML content in aquatic animals, such as crucian carp.

### 2.2. Optimization of the Extraction Conditions

The extraction conditions were compared to achieve the best extraction results in the shortest time possible. The samples were extracted using four solvents (2% *v*/*v* formic acid-acetonitrile, acetonitrile, methanol, and ethyl acetate) and two types of extraction columns (Oasis MCX SPE column and Oasis HLB SPE column). High and low VML recoveries were compared to obtain the extractant and extraction column with the best extraction effect. VML recovery rates are shown in Figure 1. VML recoveries were more suitable after extraction with 2% *v*/*v* formic acid-acetonitrile and acetonitrile using the two columns. However, the average VML recovery was greater than 100% after extraction using acetonitrile and an Oasis MCX SPE column, indicating that the MCX column had a good retention capacity for basic compounds, resulting in matrix enhancement, so the recovery rate increased. Furthermore, pigment deposition occurred when the Oasis HLB SPE column was used for extraction, and the color deepened with increasing sample volume. As a result, the solid-phase extraction column was blocked, leading to losses. Formic acid-acetonitrile was chosen for the extraction of VML from several matrices. Considering the structural similarity between VML and its metabolites, formic acid-acetonitrile was assumed to extract VML metabolites equally well. In conclusion, the 2% *v*/*v* formic acid-acetonitrile was chosen for optimum sample extraction via ultrasonic centrifugation before purification with an Oasis HLB SPE column.

### 2.3. Identification of VML Metabolites in Crucian Carp

In order to elucidate the metabolites induced by the administration of VML in carp, we employed state-of-the-art Q-TOF technology to conduct a metabolomic profiling of the liver samples collected from crucian carp, after one hour of VML injection. UPLC-Q/TOF-MS was used to analyze the protonated VML and determine the fragmentation pathways of VML. The product ion mass spectrum of VML is shown in Figure 2A. This spectrum was used as a reference to aid in the interpretation of product ions of the metabolites and to examine the high resolution and mass accuracy of the instrument. Yang et al. [8] showed that the molecular ion fracture produces a truncated fragment ion and side chain fragment ion with a mass-to-charge ratio (*m/z*) of 303.2335 and *m/z* 263.0991, respectively. The fragment ions at *m/z* 263.0991 and *m/z* 303.2335 were related to the cleavage of the ester bond from the protonated VML. The *m/z* 285.1939 was further generated from *m/z* 303.2335 after the loss of H_2_O, while *m/z* 164.0738 was generated from *m/z* 263.0991 after losing C_5_H_9_O (cleavage of the acidamide bond). Furthermore, *m/z* 147.0480 was generated from the *m/z* 164.0738 after NH_3_ loss. Notably, *m/z* 171.1495 was generated by cleaving the S-C bond in the absence of C_2_H_4_O_2_S. These results indicate the fragmentation pathways of VML. In this study, three fragmentation ions, *m/z* 263.0991, *m/z* 285.1939, and *m/z* 164.0738 were obtained in the secondary mass spectra of VML (Figure 2A). The main fragment ion at *m/z* 263.1421 was related to the cleavage of the ester bond from the protonated VML. *m/z* 303.2315 was also related to the cleavage of the ester bond, while *m/z* 285.2230 was generated from *m/z* 303.2315 after the H_2_O loss. In addition, *m/z* 164.0736 was generated from *m/z* 263.1421 after C_5_H_9_O loss (related to cleavage of the acidamide bond). The difference between the two studies could be because the collision energy was insufficient to completely fragment the ions.

The samples and the controls were first analyzed via UPLC-Q/TOF-MS. Subsequently, Metabolynx data-processing software utilizing the MDF (mass defect filtering) technique was used to identify metabolite ions, and potential metabolites were listed by the software. MS/MS experiments were further conducted to elucidate the molecular structure of the potential metabolites. The precise MS/MS spectra of these metabolites are shown in Figure 2. The chemical structures of the metabolites were determined based on chromatographic behavior, precise mass measurements, and basic rules of drug metabolism. Three VML metabolites were detected in crucian carp: V1 (580.3182, C_31_H_52_N_2_O_6_S), V2 (580.3182, C_31_H_52_N_2_O_6_S), and V3 (596.0903, C_31_H_52_N_2_O_7_S). V1 and V2 are isomers with a molecule ion of 581.3182 and a characteristic peak of 164.0651 and are related to the hydroxylation of the parent nucleus of VML. The molecule ion and characteristic peak of V3 are 597.0903 and 261.0204, respectively. V3 is related to hydroxylation of the parent nucleus of VML and the oxidation of the side chain thioether bond. Yang et al. [8] revealed that V1–V7 are the phase one metabolism products of VML in rats, chickens, and pigs. Based on this finding, we hypothesize that V1–V3 are also the phase one metabolism products of VML in crucian carp. Yang et al. [8] found 75 VML metabolites in rats and 74 VML metabolites in pigs. In this study, only three VML metabolites were found in crucian carp, which were much less abundant than those found in rats, pigs and other mammals, which is consistent with the conclusion of Kolanczyk et al. [19] that metabolites produced in fish are much less abundant than those produced in mammals.

### 2.4. Distribution of VML and Its Metabolites in Tissues

VML concentration changes in the target tissues were also analyzed (Figure 3A). VML reached maximum levels in the spleen at 0.5 h after intraperitoneal injection, which decreased with time. Moreover, VML was detected in the bile at 1 h after intraperitoneal injection, reaching maximum levels at 24 h. VML could not be completely eliminated within 96 h of injection. VML concentration peaked in other tissues, then decreased since the drug was metabolized within 24 h and completely metabolized after 72 h. Nonetheless, a large VML amount was detected in the bile at 72 h, while it was not detected in other tissues. Song et al. [18] reported that danofloxacin levels are highest in bile. Shan et al. [20] also showed that enrofloxacin (ENR) and its metabolite ciprofloxacin (CIP) form the largest area under the concentration–time curve (AUC) in the liver based on pharmacokinetics and tissue distribution and elimination analyses of ENR and its metabolite CIP in crucian carp. In this study, VML had the highest concentration in the spleen of crucian carp. This difference could be because of the different administration methods or because different drugs are metabolized in different tissues in crucian carp.

Metabolites V1, V2, and V3 are abundant in different tissues at different times (Figure 3B–D). In this study, the metabolites were mainly found in the liver at 0.1 h after VML injection. Furthermore, V1 and V2 levels were highest in the spleen and liver, while V3 level was highest only in the liver. Yang et al. [8] discovered that the amount of metabolite V1 produced in the liver microsomes of rats, chickens, swine, and goats was greater than that of V2, whereas in cows, the amount of metabolite V1 was less than that of V2. They hypothesize that the marked quantitative differences in the formation of VML metabolites among the various animal species were perhaps in relation to the levels of the involved enzymes in the liver. Therefore, we hypothesize that the differential distribution of V1, V2, and V3 levels in the liver of crucian carp may be related to the levels of relevant enzymes in the liver. The levels of the metabolites were abundant in the bile from 4 h to 12 h and in the skin after 72 h. The levels of the metabolites in the bile first increased, then decreased. The metabolism in liver was completed at 72 h. The metabolites were detected in the skin following a 72 h period, which increased with time. These results indicate that liver is the target organ of VML. The VML first acted in the liver, then circulated through the blood to other tissues, metabolized through the bile, and was finally excreted through the skin. However, the metabolites were not completely excreted at 96 h after intraperitoneal injection. Lee et al. [21] reported that *Platichthys stellatus* is highly concentrated in the kidney and liver after oral administration of plaice, *Tylosin tartrate* (TT), indicating that the kidneys and liver are crucial for drug excretion and metabolism in *P. stellatus*. Zhang et al. [22] also detected the highest concentrations of enrofloxacin (ENR) and ciprofloxacin (CIP) in the liver after multiple oral administrations, indicating that the liver is the main site of ENR metabolism in northern black flounder. In contrast, Yang et al. [23] suggested that enrofloxacin is mainly excreted through the bile in Yellow River carp, since the enrofloxacin had the largest area under the concentration–time curve (AUC) in bile. The results of this study are consistent with the study by Yang et al. VML is mainly metabolized in the bile of crucian carp, while the metabolites are mainly excreted through the skin.

### 2.5. Kinetic Investigation

The kinetic investigation of VML in crucian carp are shown in Figure 4. VML concentration in the crucian carp significantly changed after one week of VML bath and two weeks of clean water purification, with an overall trend of first increasing, then decreasing. VML concentration increased with time, reaching the enrichment balance and peaking after three days. VML concentration in the crucian carp rapidly increased on the first day, then gradually increased in the first two days, and stabilized on the third day. VML concentration decreased from days 8 and 9 after drug treatment on the seventh day. The metabolites were gradually degraded during the elimination period (after 14 days of enrichment). These results indicate that the drug reached a saturated state and purification stage in crucian carp at a certain stage, leading to its elimination.

Three VML metabolites were produced in the crucian carp during the whole enrichment and metabolism process. A large amount of metabolites was produced on the first day of the bath, on the second day, the rate of metabolite production tended to stabilize. The metabolite levels showed a fluctuating and stable state on the third to the seventh day. However, the metabolite levels decreased from the eighth day. Metabolitehe concentration increased on days 8 and 9 with decreasing prodrug concentration, indicating a maximum transformation rate. The elimination rate of the drug can be directly expressed in terms of the drug half-life (t_1/2_), as follows: t_1/2_ =0.693/k; k = −(InC_0_ − InC)/t (t_1/2_ and k represent the half-life of the drug and elimination rate constant; C_0_ and C represent the drug concentrations on the drug elimination curve, while t represents the time difference corresponding to the two concentrations).

The half-life of VML in crucian carp was 4.28 days. Sun et al. [10] found that the mean half-life of VML in Muscovy ducks (*Cairina moschata*) after intravenous (IV), intramuscular (IM) and oral administration is 2.63 h. Wang et al. [5] also found that the mean half-life of VML in broiler chickens after intravenous (IV), intramuscular (IM), and oral administration is 2.85 h. These findings indicate that the metabolism rate of VML is significantly slower in aquatic organisms, including crucian carp, than in terrestrial organisms. Furthermore, Liu et al. [24] found that the half-life (t_1/2_) of QCT in crucian carp, common carp, and grass carp are 133.97, 63.55, and 40.76 h, respectively. Furthermore, the elimination half-life (t_1/2_) was highest in crucian carp (twice and thrice that of common carp and grass carp, respectively). Therefore, the metabolic rate of VML is different in various organisms. Notably, the metabolic rate of VML is also different among different aquatic organisms.

## 3. Materials and Methods

### 3.1. Chemicals and Reagents

VML standard and Tiamulin-^13^C_4_ Fumarate (internal standard, IS) were obtained from Sigma Co. (St. Louis, MO, USA), while methanol, acetonitrile, n-hexane, and ethyl acetate (chromatographic purity) were sourced from Merck Co. (Darmstadt, Germany). Formic acid and ammonium acetate (chromatographic purity) were obtained from Sigma Co. (St. Louis, MO, USA). Ultrapure water prepared using Milli-Q pure water device (resistivity: 18.2 MΩ cm) as the water was not analyzed, but was used for fish growth.

### 3.2. Animals

A total of 120 crucian carp (weight: 175–215 g and length: 18–20 cm) were sourced from Zhejiang Zhoushan City Lincheng market. The samples were assessed using ultra-performance liquid chromatography-tandem mass spectrometry (UPLC-MS/MS), and none contained VML drug residue. Tap water treated after 72 h of aeration was used for analysis. The fish were acclimatized in a glass tank at a standardized temperature (26 ± 1 °C) and 14/10 h light/dark cycle conditions for two weeks before the experiments. The fish were fed on standard commercial diets, and the water in the glass tank was changed daily. The fish were healthy and disease-free during domestication while ensuring normal activities. The natural mortality rate was less than 5%. All animal experiments were approved by the Animal Welfare Committee of Zhejiang Marine Fisheries Research Institute (Zhoushan, Zhejiang, China) (ID Number: 2023-JG-02).

### 3.3. Sample Preparation

Blood samples were obtained from crucian carp for the analysis of pharmacokinetics and tissue distribution of VML, then transferred to a 50 mL centrifuge tube. VML internal standard solution (50 μL, 100 ng/mL) and 10 mL of 2% (*v*/*v*) formic acid-acetonitrile were added to the samples, vortexed for 30 s, sonicated for 10 min, and centrifuged at 6000 r/min for 10 min. The supernatant was then transferred to a 15 mL centrifuge tube for purification.

For other tissue samples, 2 ± 0.01 g was homogenized in a 50 mL centrifuge tube, followed by the addition of 50 μL of 100 ng/mL VML internal standard (Tiamulin-^13^C_4_) solution and 10 mL of 2% (*v*/*v*) formic acid-acetonitrile. The samples were vortexed for 30 s, ultrasonicated for 10 min, and centrifuged at 6000 r/min for 10 min. The supernatant was transferred to a 15 mL centrifuge tube for purification.

The Oasis MCX (3 mL, 60 mg) solid-phase extraction column was first activated with 3 mL methanol and equilibrated with 2% *v*/*v* formic acid water, then about 10 mL of liquid sample was extracted at 1–2 drops/s. The solid-phase extraction column was washed with 2% *v*/*v* formic acid water, methanol, and 3 mL of n-hexane. The residual liquid in the column was removed and discarded. The solution was then eluted with 4 mL of 5% ammonia-methanol solution in drops, and the eluate was collected in a 15 mL centrifuge tube. The eluent was dried with nitrogen in a 50 °C water bath. Samples were resuspended in ammonium acetate-acetonitrile (1.0 mL; 5 mmol/L, 4:1 (*v*/*v*)) solution containing 0.05% formic acid, reused after vortexing for 30 s. Finally, the supernatant was filtered through a 0.22 μm microporous cellulose membrane into an automated injection vial for metabolite identification.

### 3.4. Instrumental Conditions

UPLC-Q-TOF/MS conditions: VML and its metabolites were detected using an ACQUITY UPLC system coupled with a hybrid Q-TOF/MS (Waters, Milford, MA, USA). VML metabolites were separated using the ACQUITY ultraperformance liquid chromatography system with an Acquity UPLC BEH C18 column (50 mm × 2.1 mm, 1.7 μm particle size) (Waters, Milford, MA, USA) at 40 °C. The mobile phase (water containing 0.05% *v*/*v* formic acid and ammonium acetate 5 mmol/L (solvent A) and acetonitrile (solvent B)) was pumped at 0.3 mL/min. The gradient elution program was run as follows: 0–0.5 min, 20% solvent B; 0.5–1.5 min, 20–60% solvent B; 1.5–6.0 min, 60% solvent B; 6.0–7.0 min, 60–20% solvent B; 7.0–8.0 min, 20% solvent B (injection volume; 10 μL). Analyte was detected by MSe using electrospray (ESI) positive ion mode. LockSpray solution: leucine enkephalin (positive ion *m/z* 556.2771). Typical source conditions for maximum intensity of precursor ions were as follows: capillary voltage, 3.5 kV; source temperature, 119 °C; desolvation temperature, 385 °C; cone gas (N2) flow rate, 50 L/h; desolvation gas (N2) flow rate, 600 L/h. Data were acquired from 100 to 700 Da. Low-energy data were acquired at a collision energy of 5 eV and high-energy data using a ramped collision energy of 10−45 eV. Data processing was carried out using Metabolynx software (version 4.1).

UPLC-MS/MS conditions: Quantification of VML was performed on an Acquity UPLC unit coupled with a Quattro Premier XE mass spectrometer equipped with an electrospray ionization (ESI) source (Waters Co.). Chromatographic separation was achieved using a Waters BEH C18 UPLC column (2.1 mm × 50 mm, 1.7 μm) at 40 °C. The mobile phase (water containing 0.05% *v*/*v* formic acid and ammonium acetate 5 mmol/L (solvent A) and acetonitrile (solvent B)) was pumped at 0.3 mL/min. The gradient elution program was run as follows: 0–0.5 min, 20% solvent B; 0.5–1.5 min, 20–30% solvent B; 1.5–4.0 min, 30–40% solvent B; 4.0–4.5 min, 40–20% solvent B; 4.5–5.0 min, 20% solvent B (injection volume; 10 μL). VML was quantified using multiple reaction monitoring (MRM) in the positive ESI mode and the optimized MS parameters were set as follows: capillary voltage, 3.5 kV; source temperature, 119 °C; desolvation temperature, 385 °C; cone gas (N2) flow rate, 50 L/h; desolvation gas (N2) flow rate, 55 L/h. Data acquisition and processing were operated through Masslynx V4.1 workstation. Individual cone and collision energy voltages, as well as multiple reaction monitoring mass transitions are summarized in Table 2.

### 3.5. Method Validation

The specificity, linearity, accuracy, precision, extraction recovery, matrix effects, and stability of the method was validated using the U.S. Food and Drug Administration (FDA) Bioanalytical Method Validation [25].

The limit of detection (LOD) was defined as the concentration that produced three times the area of the signal of the baseline noise. The limit of quantification (LOQ) was defined as the concentration that produced the ratio of the area of the signal to the baseline noise of 10.

LC separation and purification conditions were optimized to avoid exogenous and endogenous interference in samples. The 5, 20, 50, 200, 500, and 1000 ng/mL were used for VML calibration due to the apparent matrix effect in plasma and various tissues. All curves were drawn using the linear regression of the mass concentration with the peak area of the quantified ions. The correlation coefficients and the slope were then calculated.

The recovery rate was calculated via the formula: R = (Amount of recovered substance/Amount of added substance) × 100%. The muscle, liver and intestine samples of three batches of crucian carp were taken, and standard solutions with five levels of 0.1, 1.0, 5.0, 10.0, and 50.0 μg/kg were added, respectively. The determination was repeated three times as described in procedure 3.3, then the recovery was averaged separately.

The samples fortified with five standards (0.1, 1.0, 5.0, 10.0, 50.0 μg/kg) were analyzed on the same day with the same instrument and by the same operator to evaluate method precision. The reproducibility (intermediate precision) results were obtained by determining samples spiked with target compounds using the same method on three separate days with the same instrument and by the same operator.

### 3.6. Pharmacokinetic and Tissue Distribution

A homogeneous VML liquid was prepared (for intraperitoneal injection) for pharmacokinetic study due to the rapid absorption, distribution, metabolism, and excretion of intraperitoneally administered drugs. After a week of culture, 60 crucian carps were randomly selected and divided into four groups, 15 crucian carps were placed in each container. Three groups were given 10 μg/mL of VML 1 mL via intraperitoneal injection, and the other group was injected with saline as the blank control group. Each individual fish was randomly selected from separate glass tanks as a replicate. The heart, liver, spleen, kidney, skin, bile, gill, intestine, gonad, muscle, and plasma tissues were removed at 0.1 h, 0.5 h, 1 h, 4 h, 12 h, 24 h, 72 h, and 96 h after administration, using anatomical tools, then frozen at −20 °C for further analysis. Each sample was well homogenized before weighing.

### 3.7. Kinetic Investigation

For kinetic analysis, a total of 60 crucian carps were randomly selected and divided into three groups, 20 crucian carps were placed in each 10 L glass tank. The drug treatment concentration of the two groups was 600 ng mL^−1^, and the other group was the blank control group without adding any drugs. The crucian carps were immersed in the VML solution for 7 days, and the exposure solution was changed every 12 h. The crucian carp were then placed in clean water for purification after seven days of bioaccumulation. The crucian carp were removed on days 1, 2, 3, 5, 7, 8, 9, 11, 14, and 21. Each individual fish was randomly selected from separate glass tanks as a replicate, and subsequently the whole crucian carp fish was homogenized for experimental processing.

## 4. Conclusions

In this study, a sensitive UPLC-MS/MS method was developed and validated to determine VML content and its three metabolites in crucian carp tissue samples. The results showed that VML first acts in the liver, then flows through the blood to other tissues, where it is metabolized in the bile. The metabolites are mainly excreted by the skin. Furthermore, the findings showed that VML has a significant transformation stage in crucian carp. The metabolite concentration increased with decreasing drug concentration on days 8 and 9. Therefore, this study provides useful data for explaining VML residues in different tissues of aquatic products, thus promoting food safety.

## Figures and Tables

**Figure 1 molecules-28-05430-f001:**
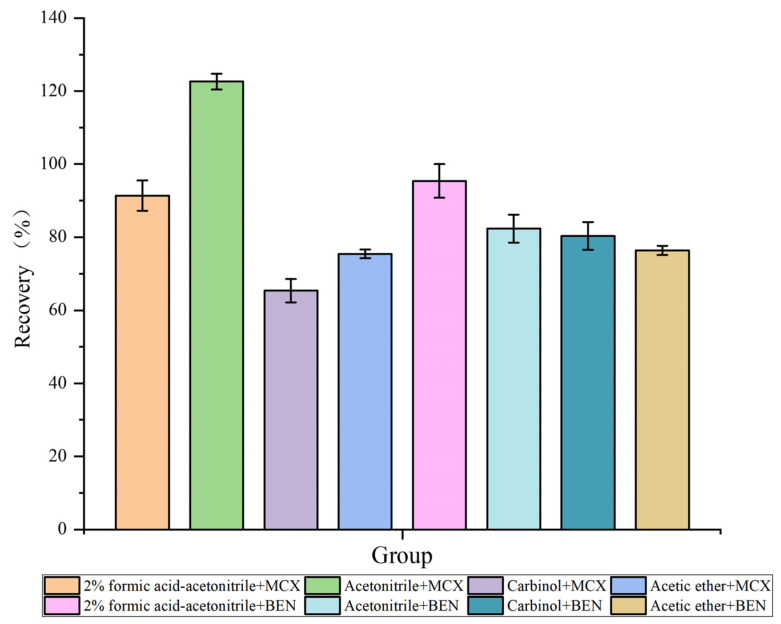
VML recoveries using different extraction columns (Oasis MCX SPE column and Oasis HLB SPE column) and extractants (2% *v*/*v* formic acid-acetonitrile, acetonitrile, methanol, and ethyl acetate) in fish samples spiked at 50 ng mL^−1^.

**Figure 2 molecules-28-05430-f002:**
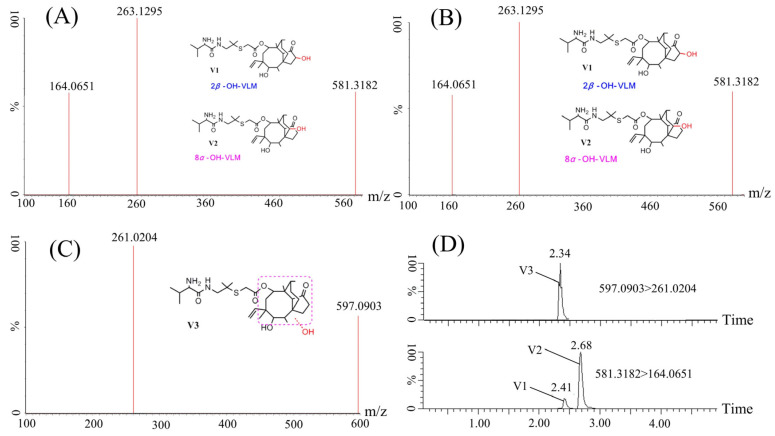
UPLC-MS/MS analysis of VML and its metabolites in crucian carp after VML injection. Mass spectra of [M + H]^+^ ions and fragmentation schemes for VML (**A**), V1 and V2 (**B**), and V3 (**C**). V1, V2, and V3 represent VML metabolites. UPLC chromatograms of VML metabolites in crucian carp (**D**).

**Figure 3 molecules-28-05430-f003:**
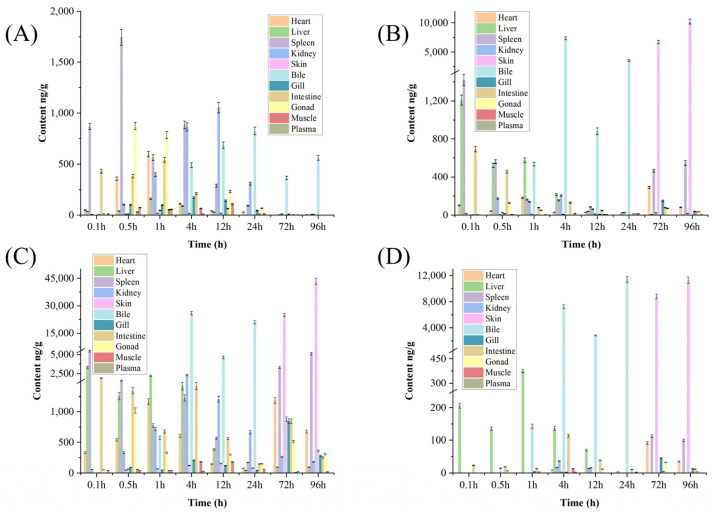
Tissue distribution of VML (**A**), V1 (**B**), V2 (**C**), and V3 (**D**). Tissue samples (heart, liver, spleen, kidney, skin, bile, gill, intestine, gonad, muscle, and plasma) were collected at different time intervals (0.1, 0.5, 1, 4, 12, 24, 72, and 96 h) after VML administration. Each point represents mean ± SD (*n* = 3).

**Figure 4 molecules-28-05430-f004:**
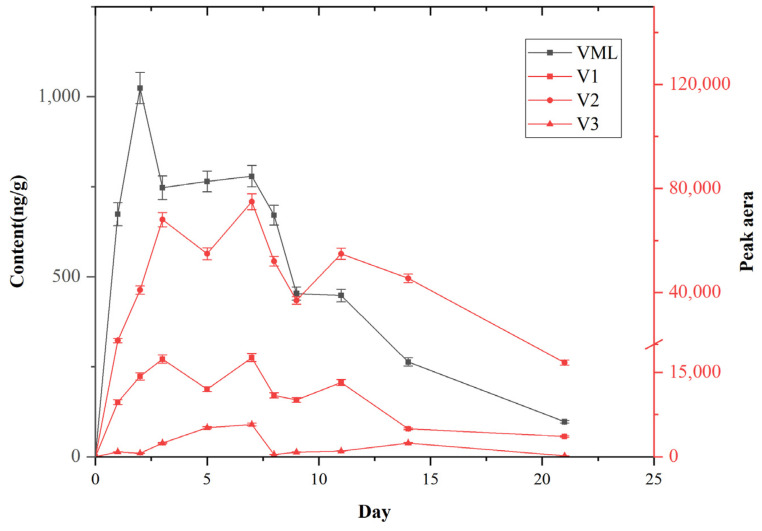
Dynamic changes in VML and its metabolites in crucian carp.

**Table 1 molecules-28-05430-t001:** Method recovery and precision (*n* = 3).

Tissues	Spiked Concentration (μg/kg)	Average Recovery Rate (%)	Intra-Day RSD (%)	Inter-Day RSD (%)
Muscle	0.1	102.63 ± 4.93	2.0	5.2
	1.0	113.78 ± 4.06	3.7	5.9
	5.0	105.30 ± 2.25	3.0	5.7
	10.0	104.77 ± 3.21	3.3	7.3
	50.0	101.42 ± 4.05	1.7	3.0
Liver	0.1	93.65 ± 4.02	6.1	7.5
	1.0	115.43 ± 5.17	5.8	7.0
	5.0	111.37 ± 4.25	3.7	6.8
	10.0	107.26 ± 4.93	5.5	6.5
	50.0	103.41 ± 4.68	4.0	5.6
Intestine	0.1	103.68 ± 3.63	6.2	7.3
	1.0	101.22 ± 2.93	4.7	5.2
	5.0	102.56 ± 4.69	4.5	6.2
	10.0	98.61 ± 3.46	3.0	5.3
	50.0	111.41 ± 5.05	6.2	7.4

**Table 2 molecules-28-05430-t002:** Analyte and internal standard transition ions and associated mass spectrometric parameters.

Analyte	Precursor Ion (*m/z*)	Product Ions (*m/z*)	Cone Voltage (V)	Collision Energy (ev)	Internal Standard
VML	565.32	263.09 ^a^ 146.80	10	1540	Tiamulin-^13^C_4_
V1	581.32	164.07	30	15	Tiamulin-^13^C_4_
V2	581.32	164.07	30	15	Tiamulin-^13^C_4_
V3	597.10	261.02	30	15	Tiamulin-^13^C_4_
Tiamulin-^13^C_4_	498.08	196.32	30	20	

Note: ^a^ Multiple reaction monitoring ions used for quantifications.

## Data Availability

The data used are confidential.

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
