# Peer review of "UPLC-MS/MS Method for Simultaneous Determination of Valnemulin and Its Metabolites in Crucian Carp: In Vivo Metabolism and Tissue Distribution Analyses"

_molecules, 2023, doi:10.3390/molecules28145430_

Round 1

Reviewer 1 Report

The work by Yang et al. seeks to study Valnemulin and its metabolites within Crucian Carp. There however appears to be some confusion within the work which hampers the understanding of the work, and this needs to be addressed to clarify what was done. This can partially be addressed though restructuring some of the sections, though I shall detail this below.

Abstract: “The concentration of the three metabolites first increased, then decreased” – This will likely need expanding as this is a very vague and generic statement.

Introduction: The introduction is likely in need of expansion, as it is very short. Some points to address include; why look for this antibiotic within aquaculture, if it is primarily used in terrestrial animals? What does it treat in aquatic animals? If it is not used for treatment (i.e., run of from farm systems) how large a problem is this in the water system? How are Crucian Carp grown (in a flow through or RAS system – see the previous point about contaminated water)?

Line 27: What are the health effects observed within humans from the listed MRLs?

Line 29: According to European – According to the European…

Line 35: I would merge “… in recent decades, with VML widely distributed…”

Line 44: Might be worth expanding why Crucian carp are important (i.e., large commercial value, consumed to a large degree in certain markets etc.).

General points regarding study design:

There appears to be confusion regarding the methodology (parts are either missing or need further clarification)

Method validation:

Line 52: the LOD and LOQ of (VML?) in (all tissues?) was 0.03…? Might be worth being more specific here. What was the MRM transition for VML, or just a single precursor ion monitored?

Line 55: ME (matrix effect?) – worth clarifying.

Optimization of the extraction conditions:

Probably worth stating that only VML was optimized, and that it was hoped that this method would be optimised for the other metabolites (as these weren’t optimised by the looks of it).

Identification of VML metabolites:

There are a few issues here, firstly there aren’t any QTOF settings (in the materials and methods), or data analysis settings/protocols here or in the method section (it looks like the settings are for the triple quad). Whilst the fragmentation of the VML structure may give a defined product ion/s, how did you filter a presumably complex QTOF data set (though the SPE would reduce this somewhat) – I think some more expansion is required, and possibly a reorder and possible combination of sections (like the following section, if compounds were elucidated by comparing control vs. VML injected)

i.e., what data tools did you use for the analysis? Did you compare QTOF data sets between VML injected and control fish (i.e., the presence and absence of compounds between the two sets of data to help ascertain the VML metabolites? – I think this might be the case (line 115), though if so, this probably should be combined with section 2.4 and expanded (with data analysis) as well as specifying the tissues used to identify metabolites in an untargeted way (i.e. data dependant acquisition of all metabolites). How were these new metabolites quantified (QTOF or triple quad MRMs – it looks like MRM from Fig. 2D)? If MRMs, what were they and to what internal standard, only VML was validated on the triple quad?

2.4 Distribution of VML: It might be worth expanding this section, for example how was VML modified within the other tissues (whilst no identification has been given for the metabolites), or likely to have been modified?

Figure 3 could be larger/clearer, as the legend is difficult to read.

Kinetic investigation: The first paragraph should be in the materials and methods section.

Line 178: only one tank containing 20 fish? Two other tanks for the control fish? I think the stats/numbers need to be made clearer (this is also a point for the later trial design) as well as the trial design (VML treatment vs. control, number of tanks per treatment and number of fish per tank). I’m also not sure what appears to be a positive control (line 180)?

Materials and Methods

Line 227: “was used for analysis” – I would change, as the water wasn’t analysed (I assume), but was used for fish growth.

Line 229: This seems to contradict line 299 (2 weeks vs 1 week), unless this one-week incubation is measured from something else (fish left to incubate post injection?).

Line 237: Is this a deuterated/heavy standard?

Line 247: “balanced” should probably be equilibrated. Also, 2% v/v (probably), I’d give v/v and w/v of solutions – this happens quite bit throughout.

Line 252: “Ammonium acetate-acetonitrile” I think you meant “samples were resuspended in ammonium acetate…”.

Line 253: mmoL/L should be mmol/L.

Instrumental conditions: The QTOF settings, data analysis should be given as these appear to be absent.

Line 261: “0.05% formic acid in hydrosolution” – probably easier to write “Water containing 0.05% v/v formic acid and ammonium acetate x% w/v”

Line 283: determination is probably calibration. I also assume the calibration was conducted to the internal standard (heavy standard?).

Line 286: I’m not sure you’re looking at accuracy, but recovery. Accuracy would be how close you are to the true value using both the calibration curve (the residuals) and possibly known values (QCs). I would just say “recovery was assessed by …”.

Line 299: crups should be carps.

Section 3.6. The stats and the number of fish should be clarified (as mentioned previously), with this section probably blended with an earlier point regarding the kinetic investigation section.

See above.

Reviewer 2 Report

Recommendation: Minor Revision

The manuscript submitted by Dr. Xiaojun Zhang, and his co-authors described “UPLC-MS/MS Method for Simultaneous Determination of Valnemulin and Its Metabolites in Crucian Carp: In Vivo Metabolism and Tissue Distribution Analyses”. Overall, the work looks significant advancement, but still some concerns need to be resolved. Therefore, I recommend this manuscript for publication after Minor revision. The following comments need to be considered before submitting.

Comments:

1. Author must improve introduction part and include novelty in the introduction

 2.Figure. 2 and 4 qualities are not good, author suggested to improve the quality.

Round 2

Reviewer 1 Report

Thanks for addressing several of my initial points, which have improved the manuscript. However, there are still quite a few issues which need to be addressed, and are somewhat confusing for the reader:

Line 20: in the skin “post” injection

Line 42: with VML is widely distributed (delete “is”).

Line 43: In present study – I think you mean “In recent studies…” also pharmacokinetics (remove s).

Line 69: Method validation – Now that I can see that the internal standard (tiamulin) was used to quantify VML and V1-V3, how did you quantify the V1-V3? Did you use the same calibration curve as VML (as presumably you don’t have standards for these metabolites?)

Line 91: Optimisation – I think you still need to say in the paper that you only optimised VML and not its metabolites. The chemical modification of VML may result in differing extraction efficiencies, and this was not explored here (for example, it can be mentioned in line 105: “formic acid-acetonitrile was chosen for the extraction of VML from several matrices and was assumed to extract VML metabolites equally well…” for example.

 Line 111: Identification of VML metabolites: whilst the materials have been updated to some extent, with some elements of data analysis, I think this still needs to be reflected in this section. I suspect (and it looks like you’ve removed the model of the mass spectrometer) that you used the same instrument to run a data dependant acquisition and then switch modes to run a TOF based MRM (rather than run a separate triple quadrupole instrument) however this is not clear. The logic of how you went from the untargeted approach to the target approach needs to be clarified.

You need to describe which tissues and time points you used to detect the metabolites (only one tissue/one time point or all tissues and time points?) You also need to state what untargeted QTOF method you used (data dependant acquisition/MSe?). This may partially explain why you found 3 metabolites vs 75 (could be time or tissue dependant). There may also be tissue specific metabolism which yields more metabolites for example (which you won’t detect if you use a targeted MRM approach.

Whilst the V1-V3 are semi-unknown, you do speculate on what the modifications are i.e., hydroxylation for example. These modifications aren’t really discussed, however. Are these common phase one metabolism products? Do you see these metabolic products from the metabolism of other antibiotics? Are there certain enzymes which are tissue specific (i.e. within the liver for V3 for example?). I think this aspect needs to be expanded.

Line 194: VML concentration in the crucian carp significantly changed… in which tissue (the blood?) or within all tissues? This needs to be specified.

Line 196: You probably should reference the figures within the text. I suspect you should mention figure 4 here.

Line 205: If this is referring to Fig 4, the metabolites appear to stabilise on day 3 (rather than day 2).

Line 247: “Blood samples were obtained…” probably should add “for kinetic analysis” as what the blood is used for is not clear. This probably should also be mentioned in line 330, as I’m assuming blood was taken during these samplings? Unless tissue was harvested?

Lines 248/254 – 2% (v/v) should be added.

Line 270 – I’m assuming you used only one mass spec system (although you’ve removed the model). The two modes (I’m assuming you used two modes, as the data analysis would suggest a untargeted DDA method of some sort. These two methods need to be clearly stated (the untargeted method, which then led to the target method).

Line 278 – Probably should be “Analytes were quantified by…”

Line 318 – “was injected with the same dose of normal saline”, probably should be “was injected with saline” as no “dosage” of saline, unless the VML solution was prepared in a specific concentration of saline?

Lines 320 and 330 – What was the n number of the sampled fish at these time points? I get around 45 fish divided by 8 (5.6, though if fish died, this could be n = 5?). Similarly, I get 40/10 for the kinetic investigation (n = 4 for each time point and n = 2 for the control). This should probably be stated in the text. Also, were individual fish used, or were fish pooled per time point for example (or were all 120 fish analysed?).

Line 330 – It should be stated what you sampled after the following sampling points, I’m assuming blood?

See above.

Author Response

Response to Reviewer 1 Comments

Point 1: Line 20: in the skin "post" injection

Response 1: Here, we have modified as suggested by the reviewer. The revised details please see page 1, line 20-21 with red mark.

Point 2: Line 42: with VML is widely distributed (delete "is").

Response 2: Thanks for your suggestions, we have modified as suggested by the reviewer. The revised details please see page 2, line 42 with red mark.

Point 3: Line 43: In present study - I think you mean "In recent studies…" also pharmacokinetics (remove s).

Response 3: Here, we have modified as suggested by the reviewer. The revised details please see page 2, line 43 with red mark.

Point 4: Line 69: Method validation - Now that I can see that the internal standard (tiamulin) was used to quantify VML and V1-V3, how did you quantify the V1-V3? Did you use the same calibration curve as VML (as presumably you don't have standards for these metabolites?)

Response 4: We appreciate your attention to detail and would like to clarify that we have indeed mentioned VML standard reference materials in the Materials section of our paper. Considering their structural similarities with VML, we have utilized VML standard reference materials to estimate the concentrations of these metabolites. Employing estimated concentrations allows us to analyze the distribution and elimination patterns of these metabolites without affecting the overall trends. Another approach we considered is directly plotting the peak area against time to analyze their distribution patterns.

Point 5: Line 91: Optimisation - I think you still need to say in the paper that you only optimised VML and not its metabolites. The chemical modification of VML may result in differing extraction efficiencies, and this was not explored here (for example, it can be mentioned in line 105: "formic acid-acetonitrile was chosen for the extraction of VML from several matrices and was assumed to extract VML metabolites equally well…" for example.

Response 5: Here, we have modified as suggested by the reviewer. The revised details please see page 4, line 104-107 with red mark.

Point 6: Line 111: Identification of VML metabolites: whilst the materials have been updated to some extent, with some elements of data analysis, I think this still needs to be reflected in this section. I suspect (and it looks like you've removed the model of the mass spectrometer) that you used the same instrument to run a data dependant acquisition and then switch modes to run a TOF based MRM (rather than run a separate triple quadrupole instrument) however this is not clear. The logic of how you went from the untargeted approach to the target approach needs to be clarified.

Response 6: We apologize for the oversight and confusion regarding the instrumental methods in our manuscript. The UPLC-MS/MS and UPLC-Q-TOF/MS instrument methods are indeed quite similar, which led us to inadvertently omit the specific conditions and confused some parameters. Here, we have rewritten Chapter 3.4 and divided it into two paragraphs to describe the instrument conditions for UPLC-MS/MS and UPLC-Q-TOF/MS, respectively.

Point 7: You need to describe which tissues and time points you used to detect the metabolites (only one tissue/one time point or all tissues and time points?) You also need to state what untargeted QTOF method you used (data dependant acquisition/MSe?). This may partially explain why you found 3 metabolites vs 75 (could be time or tissue dependant). There may also be tissue specific metabolism which yields more metabolites for example (which you won't detect if you use a targeted MRM approach.

Response 7: Thanks for the very detailed suggestions. We have modified as suggested by the reviewer. The revised details please see page 4, line 115-117, page 5, line 138-141 and page 9, line 289-304 with red mark.

Point 8: Whilst the V1-V3 are semi-unknown, you do speculate on what the modifications are i.e., hydroxylation for example. These modifications aren't really discussed, however. Are these common phase one metabolism products? Do you see these metabolic products from the metabolism of other antibiotics? Are there certain enzymes which are tissue specific (i.e. within the liver for V3 for example?). I think this aspect needs to be expanded.

Response 8: Thanks for your suggestions. We further expanded upon the aspects you mentioned in our manuscript. The revised details please see page 5, line 150-153 and page 6, line 181-188 with red mark.

Point 9: Line 194: VML concentration in the crucian carp significantly changed… in which tissue (the blood?) or within all tissues? This needs to be specified.

Response 9: Thank you for your valuable feedback on our manuscript. The kinetic investigation of crucian carp involved conducting experiments on homogenized whole fish of crucian carp. This aspect has been appropriately supplemented in the manuscript. The revised details please see page 11, line 364-366 with red mark.

Point 10: Line 196: You probably should reference the figures within the text. I suspect you should mention figure 4 here.

Response 10: Thank you for your comment and suggestion. We have included a reference to Figure 4 on page 7, line 211 of the revised manuscript.

Point 11: Line 205: If this is referring to Fig 4, the metabolites appear to stabilise on day 3 (rather than day 2).

Response 11: Thank you for your valuable feedback on our manuscript. In order to better convey our intended meaning, we have revised the sentence. The revised details please see page 8, line 223 with red mark.

Point 12: Line 247: "Blood samples were obtained…" probably should add "for kinetic analysis" as what the blood is used for is not clear. This probably should also be mentioned in line 330, as I'm assuming blood was taken during these samplings? Unless tissue was harvested?

Response 12: Thanks for your suggestions. In present study, blood samples were utilized for the analysis of pharmacokinetics and tissue distribution of VML. The kinetic investigation of crucian carp involved conducting experiments on homogenized whole fish of crucian carp. This aspect has been appropriately supplemented in the manuscript. The revised details please see page 9, line 266-267 and page 11, line 364-366 with red mark.

Point 13: Lines 248/254 - 2% (v/v) should be added.

Response 13: Here, we have modified as suggested by the reviewer.

Point 14: Line 270 - I'm assuming you used only one mass spec system (although you've removed the model). The two modes (I'm assuming you used two modes, as the data analysis would suggest a untargeted DDA method of some sort. These two methods need to be clearly stated (the untargeted method, which then led to the target method).

Response 14: We apologize for the oversight and confusion regarding the instrumental methods in our manuscript. The UPLC-MS/MS and UPLC-Q-TOF/MS instrument methods are indeed quite similar, which led us to inadvertently omit the specific conditions and confused some parameters. Here, we have rewritten Chapter 3.4 and divided it into two paragraphs to describe the instrument conditions for UPLC-MS/MS and UPLC-Q-TOF/MS, respectively.

Point 15: Line 278 - Probably should be "Analytes were quantified by…"

Response 15: Thanks for your suggestions, we have modified as suggested by the reviewer.

Point 16: Line 318 - "was injected with the same dose of normal saline", probably should be "was injected with saline" as no "dosage" of saline, unless the VML solution was prepared in a specific concentration of saline?

Response 16: Here, we have modified as suggested by the reviewer. The revised details please see page 11, line 351 with red mark.

Point 17: Lines 320 and 330 - What was the n number of the sampled fish at these time points? I get around 45 fish divided by 8 (5.6, though if fish died, this could be n = 5?). Similarly, I get 40/10 for the kinetic investigation (n = 4 for each time point and n = 2 for the control). This should probably be stated in the text. Also, were individual fish used, or were fish pooled per time point for example (or were all 120 fish analysed?).

Response 17: Thank you for your valuable feedback on our manuscript. We have modified as suggested by the reviewer. The revised details please see page 11, line 351-352 and page 11, line 364-366 with red mark.

Point 18: Line 330 - It should be stated what you sampled after the following sampling points, I'm assuming blood?

Response 18: Thanks for your suggestions. The kinetic investigation of crucian carp involved conducting experiments on homogenized whole fish of crucian carp. This aspect has been appropriately supplemented in the manuscript. The revised details please see page 11, line 364-366 with red mark.
